# PCBP2 as an intrinsic aging factor regulates the senescence of hBMSCs through the ROS-FGF2 signaling axis

**Pengbo Chen[†], Bo Li[†], Zeyu Lu[†], Qingyin Xu, Huoliang Zheng, Shengdan Jiang, Leisheng Jiang\*, Xinfeng Zheng\***

Spine Center, Xinhua Hospital, Shanghai Jiaotong University School of Medicine, Shanghai, China

## eLife assessment

In this **valuable** study, the authors aimed to identify and characterize intrinsic factors that govern the aging process of bone marrow mesenchymal stromal cells (BMSCs), which are believed to be related to osteoporosis. The authors conclude that PCBP2 is an intrinsic aging factor, the decrease of its expression during aging results in cell proliferation activity decrease and cell senescence. The study provides **convincing** evidence in support of its conclusions.

**\*For correspondence:**
jiangleisheng@xinhuamed.com.
cn (LJ);
zhengxinfeng@xinhuamed.com.
cn (XZ)

[†]These authors contributed equally to this work

**Competing interest:** The authors declare that no competing interests exist.

## Abstract

**Background:** It has been reported that loss of PCBP2 led to increased reactive oxygen species (ROS) production and accelerated cell aging. Knockdown of PCBP2 in HCT116 cells leads to significant downregulation of fibroblast growth factor 2 (FGF2). Here, we tried to elucidate the intrinsic factors and potential mechanisms of bone marrow mesenchymal stromal cells (BMSCs) aging from the interactions among PCBP2, ROS, and FGF2.

**Methods:** Unlabeled quantitative proteomics were performed to show differentially expressed proteins in the replicative senescent human bone marrow mesenchymal stromal cells (RS-hBMSCs). ROS and FGF2 were detected in the loss-and-gain cell function experiments of PCBP2. The functional recovery experiments were performed to verify whether PCBP2 regulates cell function through ROS/FGF2-dependent ways.

**Results:** PCBP2 expression was significantly lower in P10-hBMSCs. Knocking down the expression of PCBP2 inhibited the proliferation while accentuated the apoptosis and cell arrest of RS-hBMSCs. PCBP2 silence could increase the production of ROS. On the contrary, overexpression of PCBP2 increased the viability of both P3-hBMSCs and P10-hBMSCs significantly. Meanwhile, overexpression of PCBP2 led to significantly reduced expression of FGF2. Overexpression of FGF2 significantly offset the effect of PCBP2 overexpression in P10-hBMSCs, leading to decreased cell proliferation, increased apoptosis, and reduced G0/G1 phase ratio of the cells.

**Conclusions:** This study initially elucidates that PCBP2 as an intrinsic aging factor regulates the replicative senescence of hBMSCs through the ROS-FGF2 signaling axis.

**Funding:** This study was supported by the National Natural Science Foundation of China (82172474).

## Introduction

With the aging of the population, the incidence of osteoporosis is increasing globally. It is believed that the decrease in the number and 'adaptability' of bone marrow mesenchymal stromal cells (BMSCs) is one of the key factors leading to osteoporosis (*Ganguly et al., 2017*). With aging, the proliferation and function of BMSCs are impaired due to intrinsic and environmental factors (*Gibon et al., 2016*), however, the underlying mechanism remains largely unknown.

The poly(rC)-binding protein 2 (PCBP2) is an RNA-binding protein and regulates gene expression at multiple levels including mRNA metabolism and translation. It has been reported that knockout of PCBP2 led to decreased expression of p73 in a variety of cell lines, which in turn led to increased reactive oxygen species (ROS) production and accelerated cell aging (*Ren et al., 2016*). However, whether PCBP2 influences cell aging through a ROS-dependent way is still unknown. ROS is a by-product of aerobic metabolism, including superoxide anion ($O_2$-), hydrogen peroxide ($H_2O_2$), and hydroxyl radical (OH·), and has a wide range of biological targets and reactivity (*Schieber and Chandel, 2014*). The disorder of ROS affects many cell functions, such as cell proliferation, cell apoptosis, autophagy, and cellular senescence (*Schieber and Chandel, 2014*; *Luo et al., 2019*). In our pilot study, we found that the expressions of PCBP2 was significantly lower in P10-human bone marrow mesenchymal stromal cells (hBMSCs) than in P3-hBMSCs. Therefore, we speculate that PCBP2 may be one of the intrinsic factors of aging of the BMSCs and participates in the regulation of the cellular function of the cells by acting on the production of ROS.

In GSE95024 downloaded from the Gene Expression Omnibus (GEO) database, we analyzed by bioinformatics methods and found that after knockdown of PCBP2 in HCT116 cells, the expression of fibroblast growth factor 2 (FGF2) was significantly downregulated in the cells. FGF2 has been reported to have functions in cell proliferation, senescence, and G2/M arrest (*Cheng et al., 2020*; *Zhou et al., 2020*; *Salotti et al., 2013*). FGF2 was also crucial to the stemness maintenance of BMSCs with fibronectin and bone morphogenetic protein 4 (*Chen et al., 2021*). FGF2-induced inhibition of RhoA/ROCK signaling played a key role in BMSCs differentiation into endothelial cells (*Li et al., 2018*). FGF2 isoforms were also found to be able to inhibit the mineralization of BMSCs (*Xiao et al., 2013*). Meanwhile, it has been reported that ROS could mediate FGF2 release (*Kalghatgi et al., 2010*). IL-1β promoted FGF-2 expression in chondrocytes through the ROS/AMPK/p38/NF-κB signaling pathway (*Chien et al., 2016*). FGF-2 could be released by plasma-produced ROS (*Arjunan et al., 2012*).

As mentioned above, our understanding of the interactions among PCBP2, ROS, and FGF2 in the aging process of BMSCs seems to be insufficient. The in-depth exploration of the mechanism of aging of the BMSCs is expected to provide additional information for the pathogenesis and clinical intervention of osteoporosis. We hypothesize that PCBP2 is an intrinsic aging factor of BMSCs and regulates the senescence of hBMSCs through a ROS- and FGF2-dependent pathway.

## Materials and methods

### Hayflick model of cellular aging

The hBMSCs were obtained from healthy male individuals who underwent traumatic femoral or tibia shaft fracture treatment by intramedullary nailing. Cell extraction and passage were performed as previously described (*Liu et al., 2021*). hBMSCs were cultured in Dulbecco's Modified Eagle Medium (DMEM) with high glucose containing 10% fetal bovine serum. It is well accepted that the expansion of BMSCs in culture will accelerate senescence, and the differentiation potential will decrease from the sixth generation. In the tenth generation, the average number of cumulative population doublings drops from 7.7 to 1.2 (*Bonab et al., 2006*). Thus, after passage of the hBMSCs, the third-generation cells were labeled as P3 (in vitro non-senescence model, P3-hBMSCs) and the tenth-generation cells were labeled as P10 (in vitro replication senescence model, P10-hBMSCs). The cells were incubated in an incubator at 37°C with 5% $CO_2$, and the culture medium was changed every 2 days.

### GSE95024 analysis

The GSE95024 data was downloaded from the GEO database, including four groups of HCT116 cells with silenced PCBP2 and four groups of negative controls. R.3.5.2 software was used to screen out differentially expressed genes, and logFC≥2 and p<0.01 was regarded as significance. The STRING

**Table 1.** The primers used in real-time quantitative PCR (qRT-PCR).

| Gene name | Primer sequences |
|---|---|
| | Forward: 5'-ATFGTCATTTTAGCTGGATC-3' |
| PCBP2 | Reverse: 5'-GATAGATCGTGAAATGCT-3' |
| | Forward: 5'-AGAAGAGCGACCCTCACATCA-3' |
| FGF2 | Reverse: 5'-CGGTLAGCACACACTCCTTTG-3' |
| | Forward: 5'-GCTCTAGGCGGACTGTAC-3' |
| β-Actin | Reverse: 5'-CCATGCCAATGTTGTCTCTT-3' |

Three separate experiments were performed for each sample, and the $2^{-\Delta\Delta Ct}$ value was used to calculate the relative expression.

(https://string-db.org/cgi/input.pl) tool was used to predict the interaction among differentially expressed genes in protein levels.

## Unlabeled quantitative proteomics

The hBMSC were lysed with RIPA, and the protein content of the cells was quantified using BCA Protein Assay kits (Abcam, USA). Dithiothreitol (1 M), 200 µL UA buffer, 40 µL trypsin buffer, 25 µL 25 mM $NH_4HCO_3$, and 50 µL 0.1% trifluoroacetic acid (TFA) were added in sequence and then centrifuged to obtain the peptides, which were quantified using the BCA kits. After desalting with an RP-C18 solid phase extraction column, the peptides were washed with 90% acetonitrile-water containing 0.1% TFA. After elution with 90% acetonitrile-water containing 0.1% TFA, the sample was reconstituted with 0.1% formic acid in water and finally analyzed using liquid chromatography/mass spectrometry (Thermo Electron Corporation, LCQ Deca XP MAX10).

## Transfection

Cells were inoculated into 24-well plates, and when the cell confluence reached 30–50%, they were transfected with small interfering RNA (siRNA) to knock down PCBP2 (sequence: 5' -GGCCTATACCAT TCAAGGA- 3'), or transfected with plasmids overexpressing PCBP2 or FGF2. In accordance with the transfection procedure provided by GenePharma (Shanghai Gene Pharmaceutical Technology Co., Ltd.), 48 hr after transfection, the cells were used for subsequent experiments.

## Real-time quantitative PCR

The extraction of total RNA and reverse transcription were performed according to our standard protocol, as described previously (*Shi et al., 2017*; *Zheng et al., 2016*). Reverse transcription was performed using Superscript II (Thermo Fisher Scientific) and Fast SYBR Green (Thermo Fisher Scientific) was used for quantitative PCR (qPCR). The sequences of the primers used were given in *Table 1*.

## WB experiment

The steps for protein extraction and western blotting (WB) were the same as our previously published procedure (*Shi et al., 2017*). According to the WB protocol, the primary antibodies, all of which were human anti-rabbit antibodies PCBP2 (Abcam, USA), FGF2 (Abcam, USA), FRS2 (Abcam, USA), and α-tubulin (Cell Signaling Technology, CST, USA), were incubated with nitrocellulose (NC) membrane overnight at a dilution of 1:1500, and, then, incubated with secondary HRP-conjugated antibody (1:1000, rabbit anti-mouse antibody, Abcam, USA) for 1 hr. Finally, the proteins were detected using Pierce SuperSignal West Pico Chemiluminescence Detection Kits (Thermo Fisher, MA, USA), and image and protein density were photographed and calculated by ImageJ system, respectively.

## CCK-8 experiment

The cells were inoculated into 96-well plates with 2000 cells/well and cultured for 24 hr. Then, 10 µL of Cell Counting Kit-8 (CCK-8) reagent (WST-8/CCK8, ab228554, Abcam, USA) and 90 µL of serum-free medium were added to each well, and incubated for 2 hr in the 37°C, 5% $CO_2$, and humid incubator. The absorbance was measured at 450 nm, and the experiment was repeated three times.

## Detection of ROS

The cells were seeded on a black 96-well microplate with a transparent bottom at 2.5×10⁴ cells/well, and cultured overnight. The medium was then removed and 100 μL of 1× buffer was added to each well. Next, the 1× buffer was removed and 100 μL of diluted DCFDA solution added to stain the cells. The cells were incubated with diluted DCFDA solution in the dark for 45 min. The DCFDA solution was then removed, 100 μL of 1× buffer or 1× PBS added to each well, and fluorescence was measured immediately. In the presence of compound, medium or buffer, the fluorescence at Ex/Em = 485/535 nm was measured using automatic microplate reader (BioTek) end point mode.

## Flow cytometry

For apoptosis detection, after culturing the cells in six-well plates for 24 hr, the cells were collected and washed with incubation buffer, suspended with 100 μL of labeling solution, and incubated at room temperature for 10–15 min. Next, the fluorescent (SA-FLOUS) solution was added and incubated at 4°C in the dark for 20 min. Finally, the FITC fluorescence was detected at 515 nm and PI fluorescence was detected at 560 nm. Finally, samples were analyzed on FACSCalibur (BD Biosciences, San Jose, CA, USA). Data were analyzed with FlowJo software (Tree Star, Ashland, OR, USA).

For cell cycle progression, after culturing the cells in six-well plate for 24 hr, the cells were collected and fixed with 75% ethanol at 4°C for 4 hr. Next, 400 μL of CCAA solution (PI staining solution, Engreen, New Zealand) and 100 μL RNase A (100 μg/mL) was added and incubated at 4°C in the dark for 30 min. Finally, samples were analyzed on FACSCalibur (BD Biosciences, San Jose, CA, USA). The generated histogram was used to calculate the ratio of cells in the G0/G1, S, or G2/M phases.

## ROS inhibitor

5×10⁶ hBMSCs were inoculated into six-well plates, after the corresponding number of cell passages, P3-hBMSCs and P10-hBMSCs were stimulated with 2 mM ROS inhibitor (NAC, KFS289, Beijing Baiolaibo Technology Co., Ltd.)

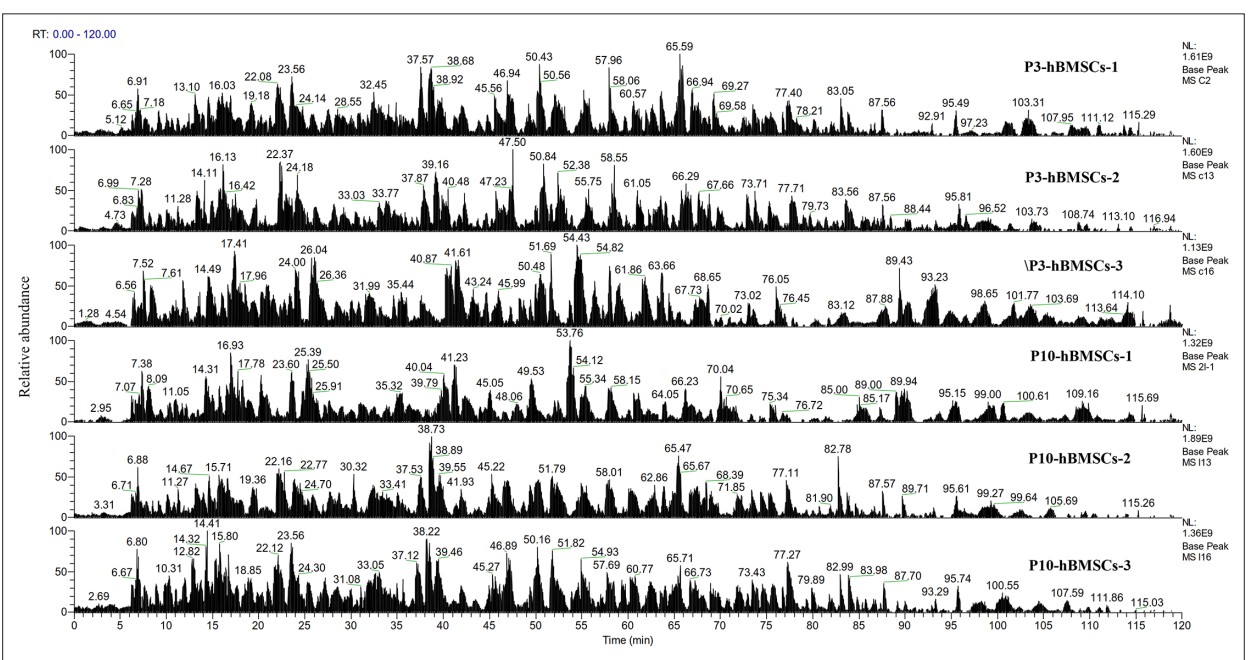

**Figure 1.** Unlabeled quantitative proteomics for the differentially expressed proteins (DEPs) between P3-human bone marrow mesenchymal stromal cells (hBMSCs) and P10-hBMSCs. The enzymolysis products were separated by capillary high-performance liquid chromatography and then analyzed by Thermo Fisher Fusion Mass Spectrometer (Thermo Fisher). The content of each peptide was expressed by detecting the intensity of positive ions, and protein analysis was performed using MaxQuant software (version number 1.6.0.1).

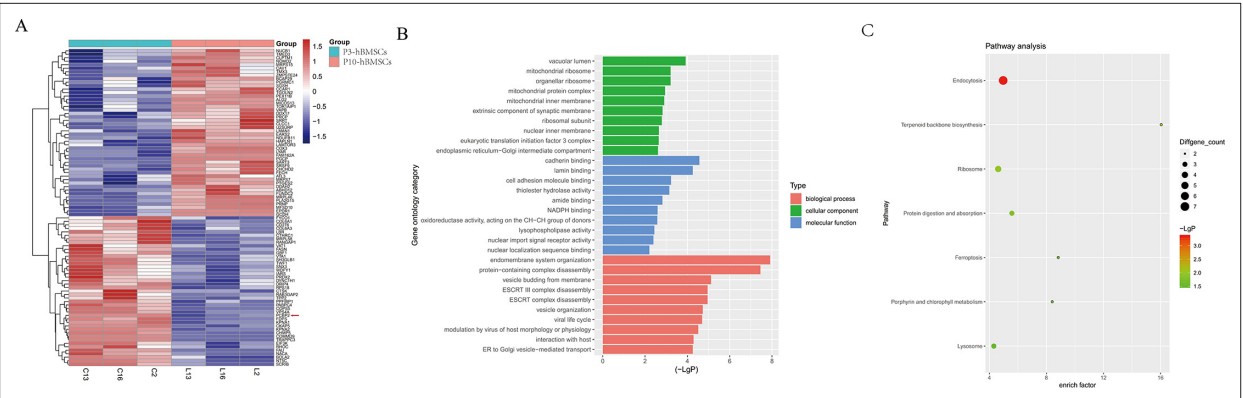

**Figure 2.** Differentially expressed proteins (DEPs) in replication senescent human bone marrow mesenchymal stromal cells (hBMSCs) and their biological functions. (**A**) A hub of 50 significantly DEPs induced by replication aging were shown as a heatmap. Gene Ontology (GO) functions (**B**) and Kyoto Encyclopedia of Genes and Genomes (KEGG) pathways (**C**) affected by these 50 DEPs. Notes: p<0.05 and logFC≥2 was regarded as indicating significance.

## Statistical analysis

Data were presented as the mean ± standard deviation (SD) for three repetitions per group. Student's t-test was used to analyze the differences between two groups, and one-way ANOVA was used to analyze the differences between multiple groups. p-Value<0.05 was considered statistically significant.

## Results

### Identification of DEPs in replicative senescent hBMSCs

In our pilot study, unlabeled quantitative proteomics was used to analyze the proteins which were differentially expressed between P3-hBMSCs and P10-hBMSCs (*Figure 1*). The sequencing results showed that 50 proteins were differentially expressed, of which 25 were significantly upregulated and 25 were significantly downregulated in the replication senescent cells (*Figure 2A*). PCBP2 was among the 25 significantly downregulated proteins (*Figure 2A*, red arrow).

Gene Ontology (GO) and Kyoto Encyclopedia of Genes and Genomes (KEGG) analyses were performed, and a variety of GO functions and KEGG signaling pathways seemed to be associated with these 50 DEPs. In GO cellular component annotations, the DEPs were significantly enriched in vacuolar lumen pathways, mitochondrial ribosome pathways, and organellar ribosome pathways. With respect to molecular function, the cadherin binding pathway, lamin binding pathway, and cell adhesion molecule binding pathway were significantly enriched in these DEPs. GO Biological Process terms included the endomembrane system organization pathway, protein-containing complex disassembly pathway, and vesicle budding from membrane pathway (*Figure 2B*). Using KEGG pathway analysis, DEPs were significantly enriched in endocytosis pathways, terpenoid backbone biosynthesis pathways, and the ferroptosis pathway (*Figure 2C*). These 50 DEPs in senescent hBMSCs seemed to affect a wide range of biological functions and signaling pathways, which might be associated with the underlying causes of cellular aging.

### Low expression of PCBP2 accentuated the characteristics of cell senescence in hBMSCs

As shown in *Figure 2A*, the expression of PCBP2, a gene downstream of the ferroptosis pathway, was significantly downregulated in P10-hBMSCs. The real-time quantitative PCR (qRT-PCR) (*Figure 3A*) and WB (*Figure 3B*) assays confirmed that the expression of PCBP2 was significantly lower in P10-hBMSCs than in P3-hBMSCs. A siRNA that knocked down PCBP2 and a plasmid that overexpressed PCBP2 were constructed, and qRT-PCR and WB were used to evaluate the efficiency of knockdown and overexpression of PCBP2 in the cells (*Figure 4*). CCK-8 results showed that after overexpressing PCBP2, the viability of both P3-hBMSCs and P10-hBMSCs increased significantly (*Figure 3C*). On the

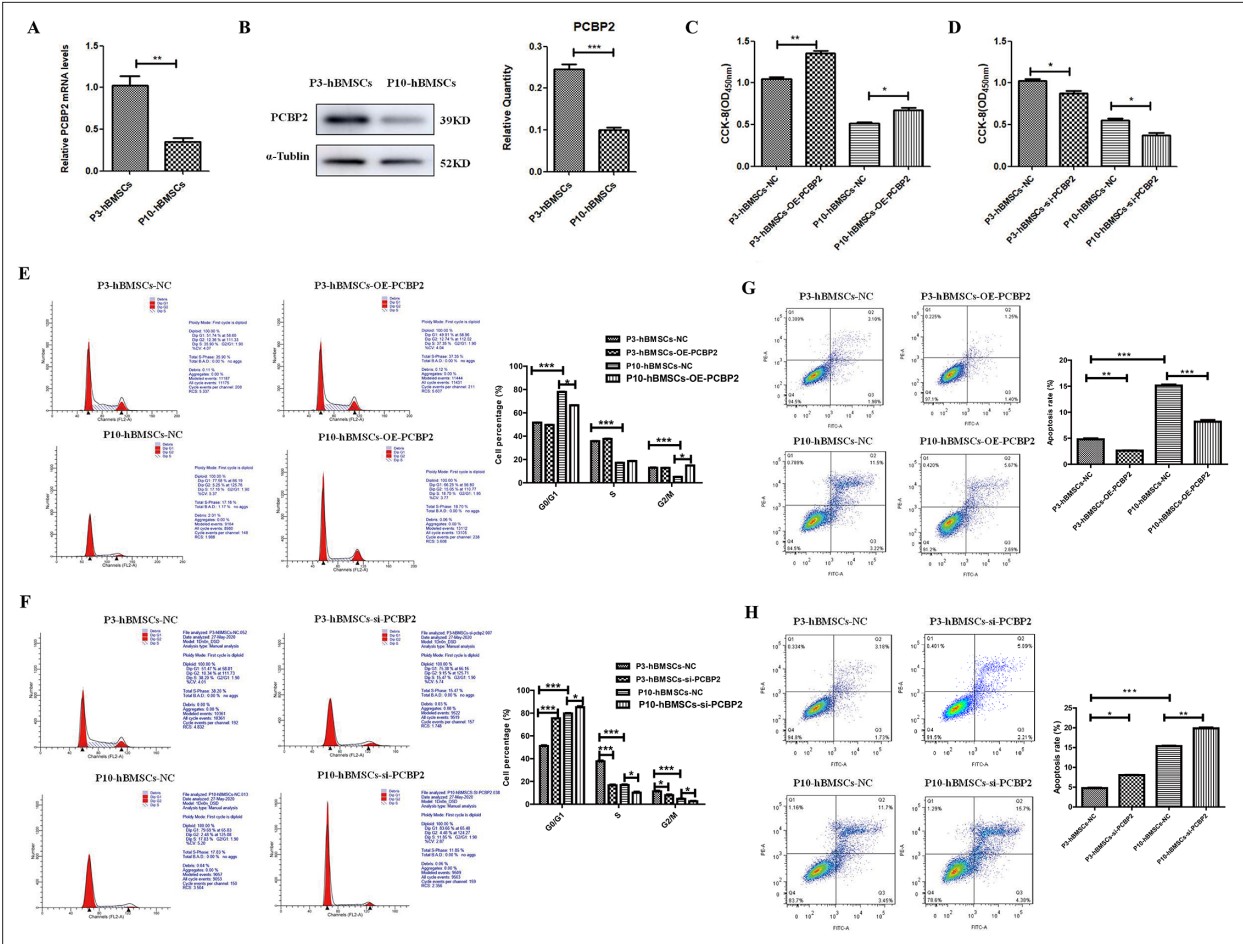

**Figure 3.** Low expression of PCBP2 accentuated the characteristics of cell senescence in human bone marrow mesenchymal stromal cells (hBMSCs). (**A**) Real-time PCR (RT-PCR) and (**B**) western blotting (WB) detection for the expression of PCBP2 in P3-hBMSCs and P10-hBMSCs. Cell Counting Kit-8 (CCK-8) assay for the effects of PCBP2 overexpression (**C**) and PCBP2 silencing (**D**) on the proliferation of P3-hBMSCs and P10-hBMSCs. Flow cytometry detection for the effects of PCBP2 overexpression (**E**) and PCBP2 silencing (**F**) on the cell cycle of P3-hBMSCs and P10-hBMSCs. The effects of PCBP2 overexpression (**G**) and PCBP2 silencing (H) on apoptosis of P3-hBMSCs and P10-hBMSCs were also detected by flow cytometry. Data were presented as mean ± SD (n=3). *p<0.05, **p<0.01, and ***p<0.001; β-actin and α-tubulin were used as the internal references for mRNA and protein detection.

The online version of this article includes the following source data for figure 3:

**Source data 1.** Original files for western blot analysis displayed in *Figure 3B*.

**Source data 2.** File containing original western blots for *Figure 3B*, indicating the relevant bands.

contrary, after silencing PCBP2, the viability of both P3-hBMSCs and P10-hBMSCs weakened significantly (*Figure 3D*).

Using flow cytometry, we found that, compared with P3-hBMSCs, P10-hBMSCs tended to remain in the G0/G1 phase of the cell cycle (*Figure 3E and F*). In P3-hBMSCs with overexpression of PCBP2, the cell cycle had no significant change. However, in P10-hBMSCs with overexpression of PCBP2, the number of cells in the G0/G1 phase decreased, while the number of cells in the mitotic phase increased significantly (*Figure 3E*). Meanwhile, in PCBP2 knocked-down P3-hBMSCs, the cell cycle was significantly blocked, with the number of cells in GO/G1 significantly increased while the numbers of cells in S phase and G2/M phase significantly decreased (*Figure 3F*). In PCBP2 knocked-down P10-hBMSCs, the cell cycle was further arrested, with the number of GO/G1 phase cells significantly increased and the number of S phase and G2/M phase cells significantly decreased (*Figure 3F*). These results demonstrated that an appropriate amount of PCBP2 could maintain the normal cell cycle, and abnormally low expression of PCBP2 would lead to arrest of the cell cycle in normal cells.

Flow cytometry also demonstrated that the cell apoptosis was substantially higher in P10-hBMSCs than in P3-hBMSCs (*Figure 3G and H*). Overexpression of PCBP2 significantly reduced apoptosis in

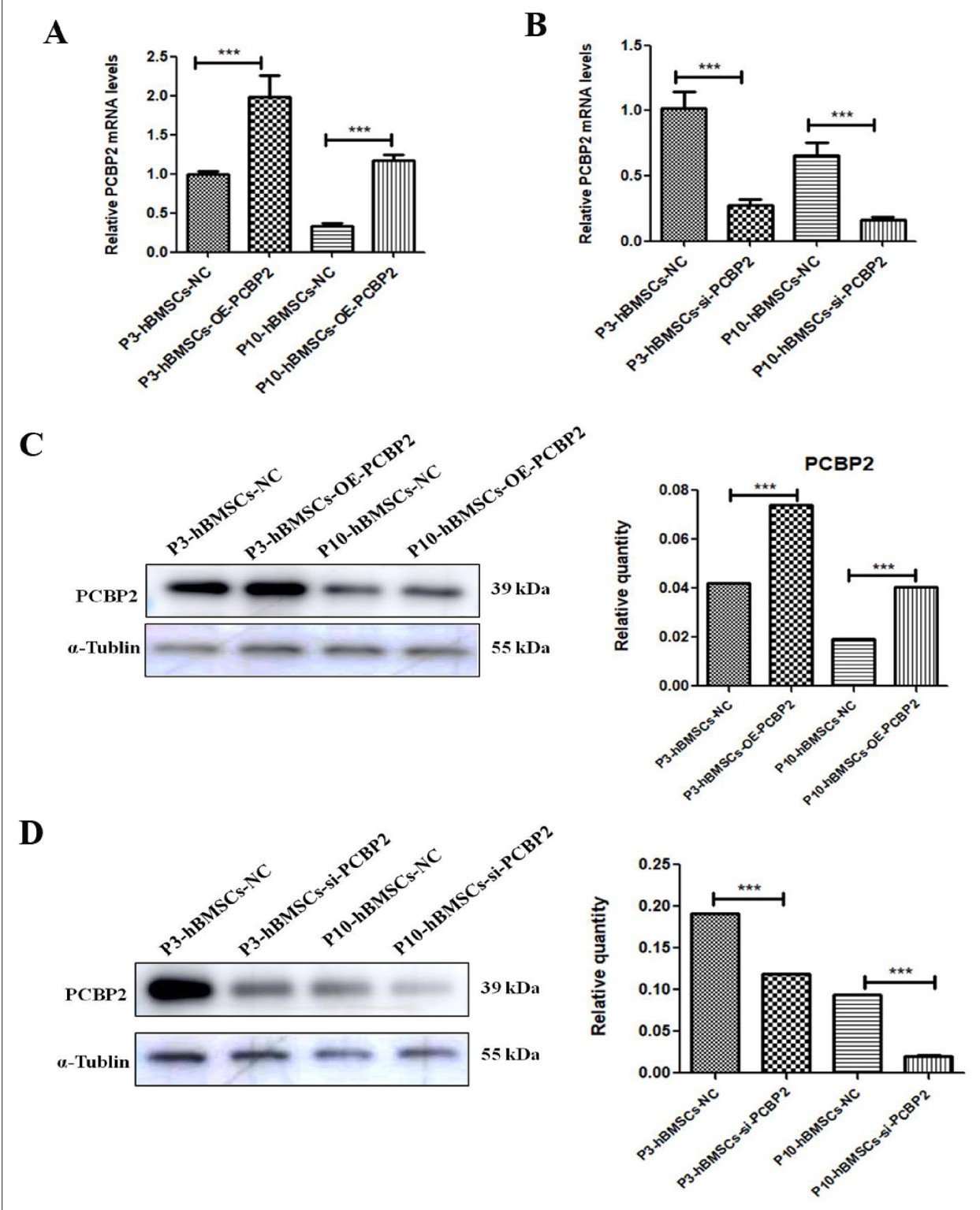

**Figure 4.** Validation of transfection effect of PCBP2. The mRNA expression of PCBP2 after transfection with plasmids that overexpressed PCBP2 (**A**) and small interfering RNA (siRNA) that knocked down PCBP2 (**B**). The effect of overexpressing PCBP2 (**C**) and knocking-down PCBP2 (**D**) as detected by western blotting (WB). Notes: Data were presented as mean ± SD (n=3). *p<0.05, **p<0.01, and ***p<0.001; β-actin and α-tubulin were used as the internal references for mRNA and proteins detection.

The online version of this article includes the following source data for figure 4:

*Figure 4 continued on next page*

*Figure 4 continued*

**Source data 1.** Original files for western blot analysis displayed in *Figure 4C*.

**Source data 2.** File containing original western blots for *Figure 4C*, indicating the relevant bands.

**Source data 3.** Original files for western blot analysis displayed in *Figure 4D*.

**Source data 4.** File containing original western blots for *Figure 4D*, indicating the relevant bands.

both P3-hBMSCs and P10-hBMSCs (*Figure 3G*), whereas silencing of PCBP2 significantly increased apoptosis in both P3-hBMSCs and P10-hBMSCs (*Figure 3H*). These results suggested that the decreased expression of PCBP2 induced by cell replicative senescence might promote the apoptosis of senescent cells.

## Low expression of PCBP2 accentuated the cell senescence of hBMSCs in a ROS-dependent way

It is well known that ROS may have important effects on the cellular functions of different cells (*Milkovic et al., 2019*). We used the functional gain-and-loss experiment to verify whether PCBP2 could influence the cellular functions of hBMSCs in a ROS-dependent pathway. The ROS detection results showed that overexpression of PCBP2 inhibited the production of ROS in both P3-hBMSCs and P10-hBMSCs (*Figure 5A*), while knockdown of PCBP2 increased the production of ROS in both P3-hBMSCs and P10-hBMSCs (*Figure 5B*). The increased ROS production caused by PCBP2 silence could be significantly rescued by adding 2 mM of antioxidant NAC to PCBP2 silenced P3-hBMSCs and P10-hBMSCs (*Figure 5C*). As shown in *Figure 3*, overexpression of PCBP2 promoted, while silencing PCBP2 inhibited the proliferation of P3-hBMSCs and P10-hBMSCs. Further, CCK-8 results showed that the antioxidant NAC could significantly reverse the decrease in cell proliferation of P3-hBMSCs and P10-hBMSCs induced by silencing PCBP2 (*Figure 5D*). Moreover, the flow cytometry result showed that compared with P3-hBMSCs+si-PCBP2 group, the apoptosis of P3-hBMSCs+si-PCBP2+NAC (2 mM) group was substantially reduced, with similar results obtained in the P10-hBMSCs (*Figure 5E*). In addition, with the introduction of NAC (2 mM) to PCBP2 silenced P3-hBMSCs and P10-hBMSCs, the proportion of cells in G0/G1 phase significantly reduced and the proportion of cells in S phase significantly increased. These results indicated that the low expression of PCBP2 induced by cell replicative senescence could inhibit cell proliferation, and induce cell apoptosis and cell arrest by increasing the production of ROS.

## Antioxidant recovered the viability of senescent hBMSCs by suppressing FGF2 expression

In GSE95024, we found that after knockdown of PCBP2, 359 genes were significantly downregulated and 332 genes were significantly upregulated (*Figure 6A*), 17 of which were significantly enriched in pathways related to the role of proteoglycans in cancer (*Figure 6B*), including 7 downregulated genes: TIMP3, PAK1, HPSE2, HSPB2, PRKACB, WNT8B, and PLAUR, and 10 upregulated genes: CAV1, FRS2, CCND1, CAV2, AKT3, SHH, FGF2, PIK3R3, WNT7A, and THBS1. Analysis of a protein interaction network generated from the STRING database, in which these genes are involved, indicates that there is a significant correlation between FGF2 and the other genes (*Figure 6C*). Therefore, we further explored the regulatory role of PCBP2 on FGF2. The qRT-PCR results showed that overexpression of PCBP2 led to significantly reduced expression of the mRNA levels of FGF2 in both P3-hBMSCs and P10-hBMSCs (*Figure 7A*). When PCBP2 was silenced, the mRNA levels of FGF2 significantly increased in both P3-hBMSCs and P10-hBMSCs (*Figure 7B*). The results of WB demonstrated that overexpression of PCBP2 led to significantly reduced protein expressions of FGF2 in both P3-hBMSCs and P10-hBMSCs (*Figure 7C*) and downregulation of PCBP2 led to a converse result (*Figure 7D*). This upregulation of FGF2 from suppression of PCBP2 could be stopped by the antioxidant NAC (*Figure 7E*).

## Low expression of PCBP2 accentuated the senescent characteristics of hBMSCs through FGF2 overexpression

Then, the functional recovery experiment was used to verify whether PCBP2 regulates cell function through FGF2. CCK-8 results showed that compared with the PCBP2 overexpression group, the cell

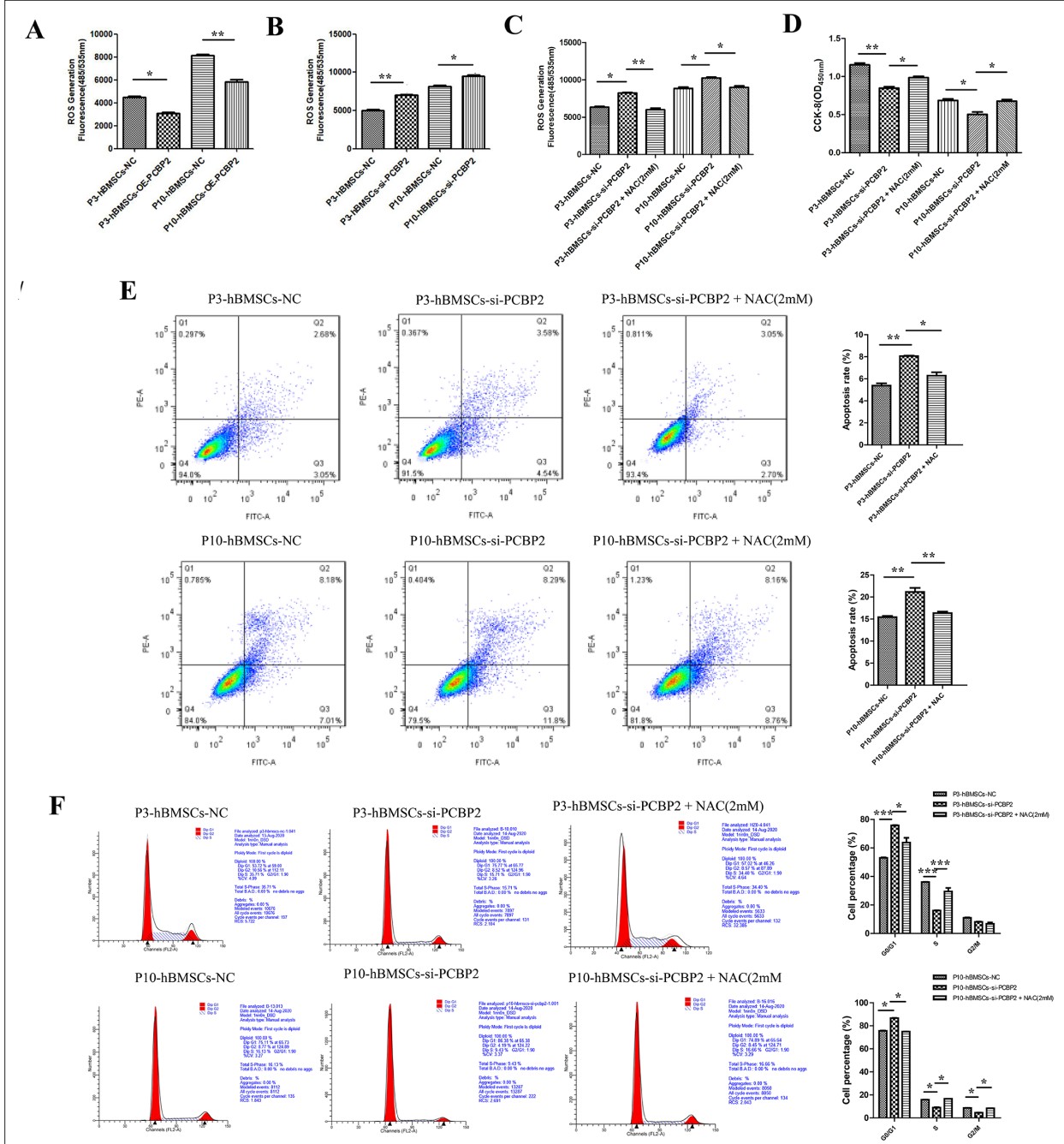

**Figure 5.** Low expression of PCBP2 inhibits cell proliferation, and promotes cell apoptosis and cell arrest in a reactive oxygen species (ROS)-dependent way. (**A**) Overexpression and (**B**) silencing of PCBP2 on ROS production in P3-human bone marrow mesenchymal stromal cells (hBMSCs) and P10-hBMSCs. (**C**) The inhibitory effect of 2 mM NAC on ROS production in P3-hBMSCs and P10-hBMSCs with silenced PCBP2. (**D**) Cell Counting Kit-8 (CCK-8) test showed that the antioxidant NAC significantly recovered the suppressed cell proliferation in P3-hBMSCs with silenced PCBP2. Flow cytometry was used to detect the effects of 2 mM NAC on the apoptosis (**E**) and cycle (**F**) of PBCP2 silenced P3-hBMSCs and P10-hBMSCs. Notes: Data were presented as mean ± SD (n=3). *p<0.05, **p<0.01, and ***p<0.001.

proliferation of P3-hBMSCs (*Figure 8A*) and P10-hBMSCs (*Figure 8B*) decreased significantly in the OE-PCBP2+OE-FGF2 group. We further used flow cytometry to detect cell apoptosis in each group. In P3-hBMSCs, overexpression of PCBP2 could significantly inhibit cell apoptosis, and overexpression of FGF2 does not significantly reverse the inhibitory effect of PCBP2 on apoptosis (*Figure 8C*). However, overexpression of FGF2 significantly reversed the apoptotic effect of PCBP2 on P10-hBMSCs (*Figure 8D*). In addition, compared with the OE-NC P3-hBMSCs and OE-NC P10-hBMSCs, the G0/

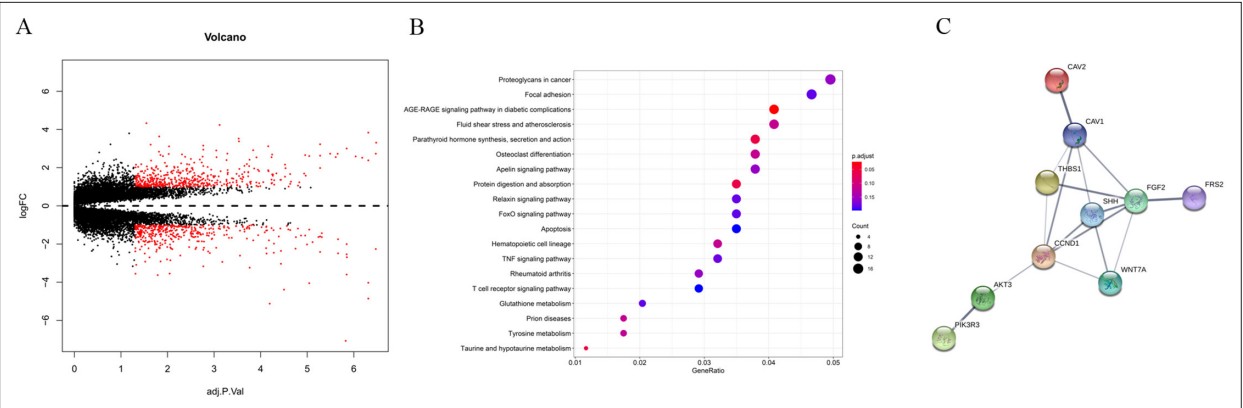

**Figure 6.** Identification of the downstream genes of PCBP2. Volcano gram of differentially expressed genes caused by PCBP2 silencing (**A**). Bubble diagram of Kyoto Encyclopedia of Genes and Genomes (KEGG) pathway analysis of differentially expressed genes (**B**). The interaction of differentially expressed proteins was predicted by STRING website.

G1 phase cell ratio was significantly reduced (*Figure 8E*) and the S phase cell ratio was significantly increased in the OE-PCBP2 group (*Figure 8F*). Moreover, compared with P3-hBMSCs (*Figure 8E*) and P10-hBMSCs (*Figure 8F*) in the OE-PCBP2 group, overexpression of FGF2 significantly reversed the PCBP2-induced cell, G0/G1 phase inhibition, and S phase promotion. In addition to G0/G1 phase and S phase, overexpression of FGF2 can significantly reverse the effect of PCBP2 on the G2/M phase of P10-hBMSCs (*Figure 8F*). These results indicated that the introduction of PCBP2 promotes cell proliferation and inhibits cell apoptosis and cell arrest by inhibiting the expression of FGF2.

## Discussion

In this work, we identified an intrinsic factor of cell aging, PCBP2, via unlabeled quantitative proteomics and characterized its biological role in the cell replicative senescence of hBMSCs in vitro. Low expression of PCBP2 in the replicative senescent hBMSCs inhibited the proliferation, promoted the apoptosis and the cell cycle arrest through a ROS-FGF2-dependent pathway.

Generally, cell senescence is regarded as an irreversible cell cycle arrest, which is considered to be an evolutionary process established and maintained by the p16 $^{INK4A}$ and/or p53-p21 pathway (*Cheng et al., 2017*). However, with the suppression of p53 expression, senescent mouse embryonic fibroblasts rapidly re-entered the cell cycle (*Dirac and Bernards, 2003*), which inspired us with new understanding of cell senescence. The phenotype of cellular senescence is triggered by a variety of senescence stressors, which in turn affect multiple signaling pathways and gene expressions (*Cheng et al., 2017*). In order to validate the results of our pilot study, we used qRT-PCR and WB analysis and confirmed that both the mRNA and protein expressions of PCBP2 were substantially lower in the senescent P10-hBMSCs than in the non-senescent P3-hBMSCs. Therefore, PCBP2 should be an intrinsic factor for cell aging and participate in the senescence of hBMSCs.

Cell cycle arrest, loss of cell proliferation, and increase of cell apoptosis are the main characteristics of cell senescence (*Dodig et al., 2019*). In present study, we found that PCBP2 silencing significantly promoted the retention of hBMSCs in the G0/G1 phase and significantly reduced the proportion of cells in the S and G2/M phases. These results are consistent with previous reports that in MEFs, the loss of PCBP2 leads to the decrease of p73 expression, which in turn leads to the acceleration of cell senescence (*Ren et al., 2016*). Meanwhile, we found that the apoptosis of replicative senescent hBMSCs increased after PCBP2 was silenced, while decreased after PCBP2 was overexpressed. After multiple proliferation and division of cells, the telomeres shorten, the cells reach the limit of normal cell division and turn into replicative senescence. This process is also known as Hayflick limit (*Liu et al., 2019*). Overexpression of PCBP2 in the P10-hBMSCs promoted the proliferation of the replicative senescent cells, suggesting that PCBP2 might have the ability to extend the Hayflick limit or waken the cells from replicative senescence. Further studies are needed to verify these findings.

Cell senescence can be induced by many factors, such as oxidative stress, cell culture and telomere shortening, and disturbances in mitochondrial homeostasis (*Regulski, 2017*). Abnormal mitochondrial

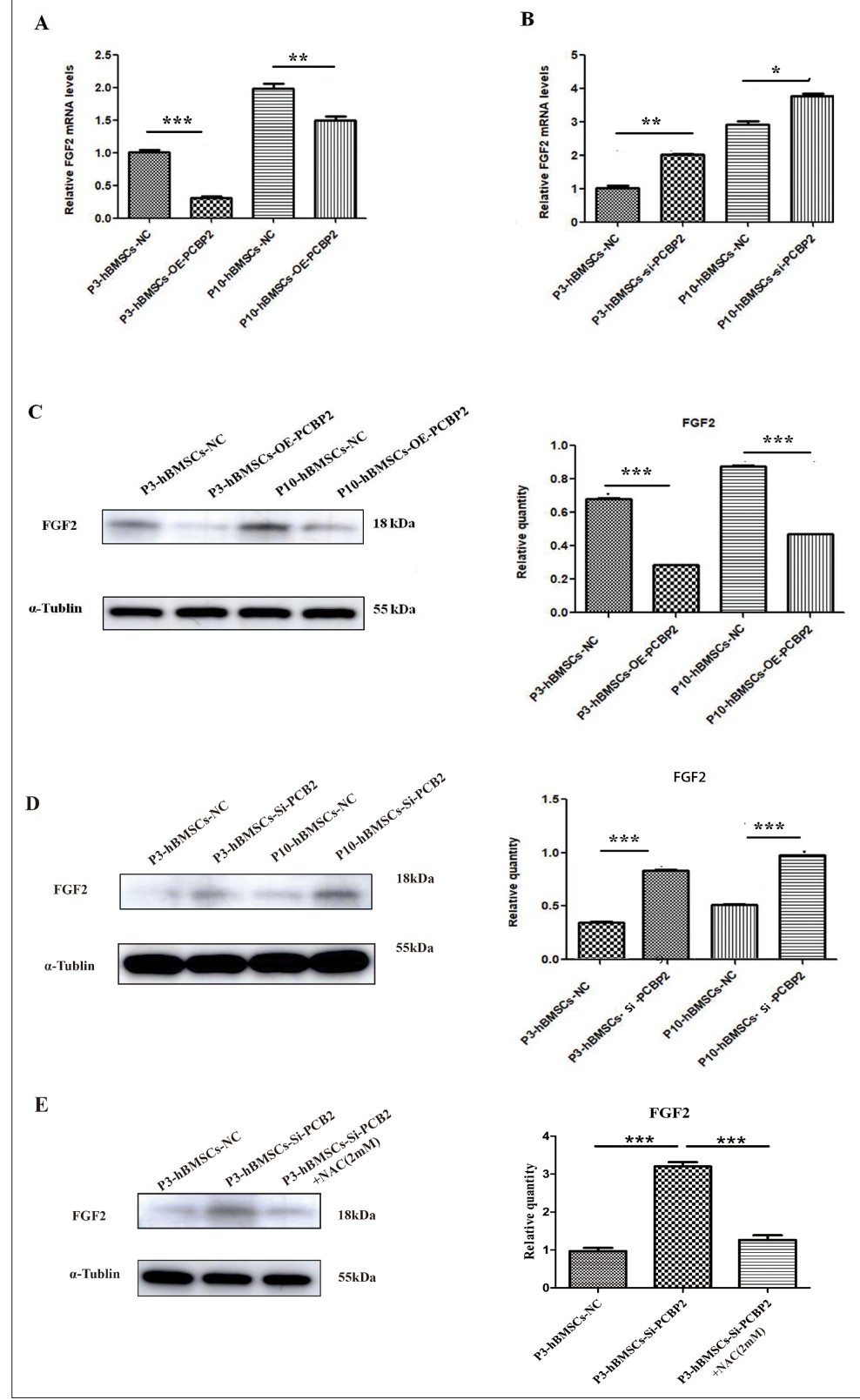

**Figure 7.** PCBP2 inhibits the expression of fibroblast growth factor 2 (FGF2) in P3-human bone marrow mesenchymal stromal cells (hBMSCs) and P10-hBMSCs. The regulation of FGF2 mRNA levels by overexpression (**A**) and silencing (**B**) of PCBP2 was detected by real-time PCR (RT-PCR) and was confirmed in protein levels via western blotting (WB) assay (**C, D**). (**E**) Upregulation of FGF2 protein level in P3-hBMSCs with silenced PCBP2 was stopped

*Figure 7 continued on next page*

*Figure 7 continued*

by 2 mM NAC. Notes: Data were presented as mean ± SD (n=3). *p<0.05, **p<0.01, and ***p<0.001; β-actin and α-tubulin were used as the internal references for mRNA and proteins detection.

The online version of this article includes the following source data for figure 7:

**Source data 1.** Original files for western blot analysis displayed in *Figure 7C*.

**Source data 2.** File containing original western blots for *Figure 7C*, indicating the relevant bands.

**Source data 3.** Original files for western blot analysis displayed in *Figure 7D*.

**Source data 4.** File containing original western blots for *Figure 7D*, indicating the relevant bands.

**Source data 5.** Original files for western blot analysis displayed in *Figure 7E*.

**Source data 6.** File containing original western blots for *Figure 7E*, indicating the relevant bands.

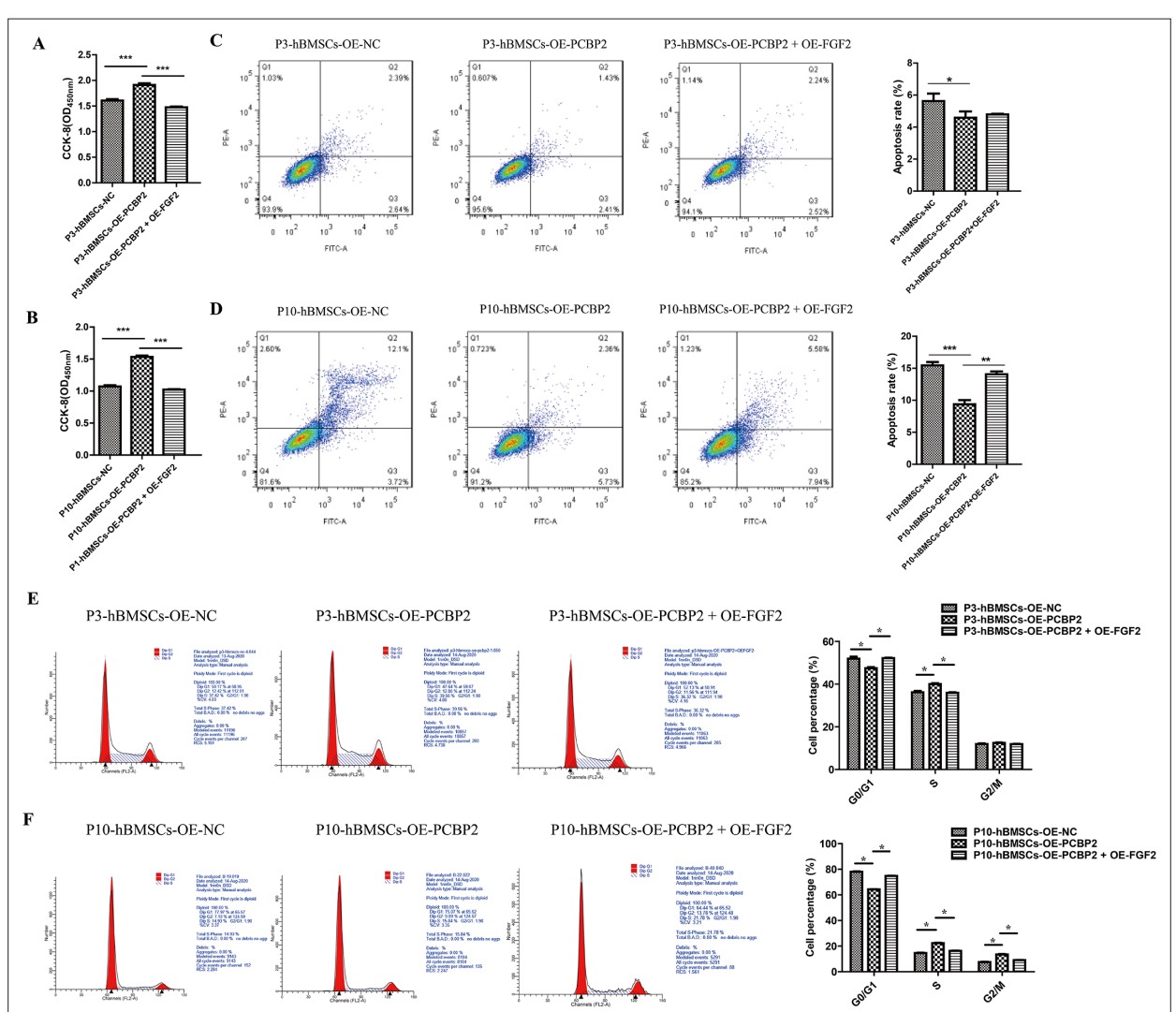

**Figure 8.** Overexpression of PCBP2 promotes cell proliferation, and inhibits cell apoptosis and cell arrest in a fibroblast growth factor 2 (FGF2)-dependent way. Cell Counting Kit-8 (CCK-8) detects the effect of overexpression of PCBP2 and FGF2 on the cell proliferation of P3-human bone marrow mesenchymal stromal cells (hBMSCs) (**A**) and P10-hBMSCs (**B**). Flow cytometry was used to detect the apoptosis of P3-hBMSCs (**C**) and P10-hBMSCs (**D**) in NC, OE-PCBP2, and OE-PCBP2+OE-FGF2 groups. In addition, in NC group, OE-PCBP2 group and OE-PCBP2+OE-PCBP2 group, the cell cycle of P3-hBMSCs (**E**) and P10-hBMSCs (**F**) were detected by flow cytometry. Notes: Data were presented as mean ± SD (n=3). *p<0.05, **p<0.01, and ***p<0.001.

homeostasis leads to an increase in the production of ROS (*Victorelli and Passos, 2019*). Meanwhile, mitochondrial ROS aggravate cell senescence by enhancing DNA damage (*Passos et al., 2010*), which in turn forms the characteristic phenotype of senescent cells. *Ren et al., 2016*, reported that PCBP2 could inhibit the production of ROS by regulating the mRNA stability of p73 in MEFs. By interfering with the expression of PCBP2, we investigated the role of PCBP2 on the production of ROS in hBMSCs and obtained similar results. It is reported that PCBP2 is a versatile adaptor protein that binds iron and delivers it to ferritin for storage (*Frey et al., 2014*). The main components of iron are stored in mitochondria, lysosomes, cytosol, and nucleus, at concentrations of ~16 μM, ~16 μM, ~6 μM, and ~7 μM, respectively (*Paul et al., 2017*). Iron overload at the cellular level leads to an increase in ROS production and results in oxidative stress (*Andrews, 1999*). However, the expression of PCBP2 was low in the replicative senescent hBMSCs, suggesting that the increase of ROS production in the senescent cells with low expression of PCBP2 might not depend on the transfer of iron. Future studies on the regulation mechanism of PCBP2 on ROS production are needed.

We found that downregulating the expression of PCBP2 in hBMSCs could increase the ROS production in the cells. ROS are involved in regulating the activity of multiple signaling pathways, which are related to cell growth, cell apoptosis, and the cell cycle (*Zheng et al., 2018*). Moderate level of ROS initiates the differentiation of stem cells, while high level of ROS leads to cell senescence and cell death of stem cells (*Bigarella et al., 2014*). Through functional recovery experiments, we found that overexpression of PCBP2 promoted the proliferation and inhibited the apoptosis and cell cycle arrest of P10-hBMSCs in a ROS-dependent way. This result not only challenges the traditional theory that cell senescence is irreversible cell cycle arrest (*de Magalhães and Passos, 2018*), but also indicates that the replicative senescent hBMSCs can be wakened in some extent by interventions on the production of ROS.

As mentioned in the Introduction, FGF-2 can be released by ROS (*Kalghatgi et al., 2010*; *Chien et al., 2016*; *Arjunan et al., 2012*). Our experiment also demonstrated that the increase of FGF2 induced by decreased PCBP2 expression can be reversed by oxidative stress inhibitors. Therefore, we concluded that PCBP2 inhibits ROS expression to a certain extent, and low ROS expression reduced FGF2 production, thus, regulating the aging of hBMSCs. FGF2 promotes catabolism and subsequent anabolism by specifically binding to FGFR1 (*Yan et al., 2011*) and is involved in regulating multiple downstream signal pathways, including PI3K/AKT, STAT1/p21, and RAS/MAPK kinase pathways (*Teven et al., 2014*). Therefore, FGF2 plays an important role in regulating cell function. Through sequencing analysis, we found that when PCBP2 was knocked down, the expression of FGF2 was significantly increased, which was confirmed by qRT-PCR and WB. Combined with the functional recovery experiment, we found that overexpression of PCBP2 in the P10-hBMSCs could promote cell proliferation and make the cells re-enter the S phase and G2/M phase, while inhibiting the apoptosis of the cells by inhibiting FGF2. However, whether FGF2 mediates the cellular functions of hBMSCs through PI3K/AKT, STAT1/p21, and RAS/MAPK kinase pathways remains to be confirmed.

In conclusion, this study initially elucidates that PCBP2 as an intrinsic aging factor regulates the replicative senescence of hBMSCs through the ROS-FGF2 signaling axis. However, how does FGF2 interact with ROS in hBMSCs? In addition to FGF2 and ROS, does PCBP2 have other potential mechanisms for the aging of hBMSCs? These questions remain unanswered in future research.

## Acknowledgements

This study was supported by the National Natural Science Foundation of China (82172474).

## Additional information

### Funding

| Funder | Grant reference number | Author |
| --- | --- | --- |
| National Natural Science Foundation of China | 82172474 | Leisheng Jiang |

| Funder | Grant reference number | Author |
|---|---|---|

The funders had no role in study design, data collection and interpretation, or the decision to submit the work for publication.

## Author contributions

Pengbo Chen, Writing – original draft; Bo Li, Data curation; Zeyu Lu, Data curation, Software; Qingyin Xu, Formal analysis, Methodology; Huoliang Zheng, Conceptualization, Resources; Shengdan Jiang, Conceptualization, Supervision; Leisheng Jiang, Writing – review and editing; Xinfeng Zheng, Conceptualization, Writing – review and editing

## Author ORCIDs

Zeyu Lu ⓘ https://orcid.org/0000-0002-3051-9183
Xinfeng Zheng ⓘ https://orcid.org/0000-0003-0837-785X

## Ethics

The experimental protocol adhered to the ethical principles outlined in the Helsinki Declaration, including respect for individuals, beneficence, non maleficence, and justice. The study was approved by the Ethics Committee of Xinhua Hospital Affiliated to Shanghai Jiao Tong University School of Medicine (XHECNSFC2021159). Human bone marrow derived mesenchymal stem cells (BMSCs) were used with the patient's informed consent, and consent for publication of this article was also obtained from the patient.

Reviewer #1 (Public Review): https://doi.org/10.7554/eLife.92419.2.sa1
Reviewer #2 (Public Review): https://doi.org/10.7554/eLife.92419.2.sa2
Author response https://doi.org/10.7554/eLife.92419.2.sa3

---

# Additional files

## Supplementary files

MDAR checklist

Source data 1. Protein identification quantitative summary of *Figure 1* and *Figure 2*.

## Data availability

The datasets used and/or analyzed during the current study are available on Dryad and can be accessed through the https://doi.org/10.5061/dryad.hx3ffbgqk.

The following dataset was generated:

| Author(s) | Year | Dataset title | Dataset URL | Database and Identifier |
|---|---|---|---|---|
| Zheng X | 2025 | hBMSCs Protein profile | https://doi.org/10.5061/dryad.hx3ffbgqk | Dryad Digital Repository, 10.5061/dryad.hx3ffbgqk |

The following previously published dataset was used:

| Author(s) | Year | Dataset title | Dataset URL | Database and Identifier |
|---|---|---|---|---|
| Zaccara S | 2020 | p53-induced apoptosis is specified by a translation program regulated by PCBP2 and DHX30 | https://www.ncbi.nlm.nih.gov/geo/query/acc.cgi?acc=GSE95024 | NCBI Gene Expression Omnibus, GSE95024 |

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
