## [Editor Report · eLife assessment]

In this **valuable** study, the authors aimed to identify and characterize intrinsic factors that govern the aging process of bone marrow mesenchymal stromal cells (BMSCs), which are believed to be related to osteoporosis. The authors conclude that PCBP2 is an intrinsic aging factor, the decrease of its expression during aging results in cell proliferation activity decrease and cell senescence. The study provides **convincing** evidence in support of its conclusions.

---

## [Referee Report · Reviewer #1 (Public Review)]

Summary:

This study aimed to elucidate the intrinsic factors and potential mechanisms of BMSCs aging from the interactions among PCBP2, ROS, and FGF2. This study represents the first study to reveal PCBP2 as an intrinsic aging factor to regulate the replicative senescence of hBMSCs through ROS-FGF2 signaling. This study provides convincing evidence to support the above conclusion.

Strengths:

This study utilized multiple in vitro approaches, such as proteomics, siRNA, and overexpression, to demonstrate that PCBP2 is an intrinsic factor of BMSC aging.

Weaknesses:

This study did not perform in vivo experiments.

---

## [Referee Report · Reviewer #2 (Public Review)]

Summary:

The authors were trying to identify and characterize the intrinsic factors that control the process of cell aging of bone marrow mesenchymal stromal cells (BMSCs), which is believed to be related to osteoporosis.

Strengths:

The method is reasonable. The concept and methods used in this work can be easily extended to other systems and cells to study their aging process. It is also interesting to further examine if PCBP2 functions as an intrinsic aging factor in these other cell types.

The results are solid, supporting the claims and conclusions. The authors successfully identified and characterized PCBP2 as one of the intrinsic aging factors for BMSC cells.

Weaknesses:

It is unclear if PCBP2 can also function as an intrinsic factor for BMSC cells in female individuals. More work may be needed to further dissect the mechanism of how PCBP2 impacts FGF2 expression. Could PCBP2 impact the FGF2 expression independent of ROS?

Additional context that would help readers interpret or understand the significance of the work:

In the current work, the authors studied the aging process of BMSC cells, which are related to osteoporosis. Aging processes also impact many other cell types and their function, such as in muscle, skin, and the brain.

---

## [Author Response]

**Reviewer #1 (Public Review):**
[...] Strengths:This study utilized multiple in vitro approaches, such as proteomics, siRNA, and overexpression, to demonstrate that PCBP2 is an intrinsic factor of BMSC aging.Weaknesses:This study did not perform in vivo experiments.

Response: We will continue to conduct animal experiments in subsequent studies.

**Reviewer #2 (Public Review):**
[...] Weaknesses:It is unclear if PCBP2 can also function as an intrinsic factor for BMSC cells in female individuals. More work may be needed to further dissect the mechanism of how PCBP2 impacts FGF2 expression. Could PCBP2 impact the FGF2 expression independent of ROS?

Response: Thank you very much for your valuable comments, which is also the focus of our follow-up work. We will sort out the data and publish the relevant research results as soon as possible.

Additional context that would help readers interpret or understand the significance of the work: In the current work, the authors studied the aging process of BMSC cells, which are related to osteoporosis. Aging processes also impact many other cell types and their function, such as in muscle, skin, and the brain.

Response: Thank you very much for your valuable comments, we will continue to improve the writing logic of the article to make the article more understandable.